# Meroterpenoids from *Gongolaria abies-marina* against Kinetoplastids: *In Vitro* Activity and Programmed Cell Death Study

**DOI:** 10.3390/ph16040476

**Published:** 2023-03-23

**Authors:** Desirée San Nicolás-Hernández, Rubén L. Rodríguez-Expósito, Atteneri López-Arencibia, Carlos J. Bethencourt-Estrella, Ines Sifaoui, Lizbeth Salazar-Villatoro, Maritza Omaña-Molina, José J. Fernández, Ana R. Díaz-Marrero, José E. Piñero, Jacob Lorenzo-Morales

**Affiliations:** 1Instituto Universitario de Enfermedades Tropicales y Salud Pública de Canarias, Universidad de La Laguna, Avda. Astrofísico Fco. Sánchez, S/N, 38206 La Laguna, Tenerife, Spain; 2Departamento de Obstetricia y Ginecología, Pediatría, Medicina Preventiva y Salud Pública, Toxicología, Medicina Legal y Forense y Parasitología, Universidad de La Laguna, 38206 La Laguna, Tenerife, Spain; 3Red de Investigación Colaborativa en Enfermedades Tropicales (RICET), Instituto de Salud Carlos III, 28006 Madrid, Spain; 4Departamento de Infectómica y Patogénesis Molecular, Centro de Investigación y de Estudios Avanzados del Instituto Politécnico Nacional, Ciudad de Mexico 07360, Mexico; 5Facultad de Estudios Superiores Iztacala, Medicina, Universidad Nacional Autónoma de México (UNAM), Tlalnepantla 54090, Mexico; 6Instituto Universitario de Bio-Orgánica Antonio González, Departamento de Química Orgánica, Universidad de La Laguna, Avda. Astrofísico Fco. Sánchez 3, 38206 La Laguna, Tenerife, Spain; 7Instituto de Productos Naturales y Agrobiología, Consejo Superior de Investigaciones Científicas, Avda. Astrofísico Fco. Sánchez 3, 38206 La Laguna, Tenerife, Spain; 8Consorcio Centro de Investigación Biomédica En Red (CIBER) de Enfermedades Infecciosas (CIBERINFEC), Instituto de Salud Carlos III, 28006 Madrid, Spain

**Keywords:** *Gongolaria abies-marina*, *Leishmania*, *Trypanosoma cruzi*, chemotherapy, meroterpenoids, apoptosis-like, autophagy, marine natural products

## Abstract

Leishmaniasis and Chagas disease affect millions of people worldwide. The available treatments against these parasitic diseases are limited and display multiple undesired effects. The brown alga belonging to the genus *Gongolaria* has been previously reported as a source of compounds with different biological activities. In a recent study from our group, *Gongolaria abies-marine* was proven to present antiamebic activity. Hence, this brown alga could be a promising source of interesting molecules for the development of new antiprotozoal drugs. In this study, four meroterpenoids were isolated and purified from a dichloromethane/ethyl acetate crude extract through a bioguided fractionation process targeting kinetoplastids. Moreover, the in vitro activity and toxicity were evaluated, and the induction of programmed cell death was checked in the most active and less toxic compounds, namely gongolarone B (**2**), 6*Z*-1′-methoxyamentadione (**3**) and 1′-methoxyamentadione (**4**). These meroterpenoids triggered mitochondrial malfunction, oxidative stress, chromatin condensation and alterations of the tubulin network. Furthermore, a transmission electron microscopy (TEM) image analysis showed that meroterpenoids (**2**–**4**) induced the formation of autophagy vacuoles and ER and Golgi complex disorganization. The obtained results demonstrated that the mechanisms of action at the cellular level of these compounds were able to induce autophagy as well as an apoptosis-like process in the treated parasites.

## 1. Introduction

Leishmaniasis and Chagas disease are considered by the World Health Organization (WHO, Geneva, Switzerland) as neglected diseases that affect more than 1 billion people. These diseases mainly affect people who live in the poorest countries in tropical and subtropical climates [1]. Leishmaniasis is globally distributed and there are around 700,000 to 1 million new cases per year around the world. Climate change, socioeconomic conditions, environmental changes, and malnutrition are involved in the epidemiology and development of this disease [2]. Leishmaniasis is caused by more than 20 different species of *Leishmania* and it is transmitted by the bite of infected female phlebotomine sandflies. Moreover, the vector carries the mobile form of parasite life cycle (promastigote), whereas the infected host cells harbor the immobile and intracellular form (amastigotes). In addition, the disease manifest in three main clinical scenarios: cutaneous leishmaniasis (the most common form of the disease, causing skin lesions and ulcers), mucocutaneous leishmaniasis (characterized by partial or total destruction of mucous membranes of the nose, mouth and throat) and visceral leishmaniasis (liver, spleen, and bone marrow could be damaged by the parasite and is fatal in over 95% of cases) [2].

On the other hand, Chagas disease, also known as American trypanosomiasis, is found mainly in Latin America, with this region being the main focus of transmission. However, in the last decades, the number of cases has increased in the United States of America (USA) and Canada, as well as European, African and Western Pacific countries due to population mobility, urbanization and immigration. Hence, an estimated number of 75 million people are at risk of infection [3]. Chagas disease is caused by the protozoan parasite *Trypanosoma cruzi* and is transmitted by the contact with feces from infected blood-sucking triatomine bugs, the vector of this pathogen. Metacyclic trypomastigotes (the mobile form of the parasite) present in the feces of triatomine bugs penetrate through wounds in the skin or mucous membrane of the mammalian host and invade surrounding cells (where they transform into amastigotes). The amastigotes are able to multiply in the cytosol and transform into trypomastigotes, which are released after cell lysis, infecting neighboring cells and migrating to different tissues. Furthermore, it could be also transmitted by oral and congenital transmission, organ transplantation and blood transfusions from infected donors and laboratory accidents [4]. American trypanosomiasis is characterized by two phases of the disease: acute and chronic phases. The acute phase is absent in most cases. However, people who manifest symptoms used to present skin lesions or a purplish swelling of the lids of one eye (called *Romaña* sign), fever and headache. Cardiac and digestive disorders are characteristics in the chronic phase, which can lead to the death of the patient due to heart arrhythmia or heart failure [2].

Both leishmaniasis and Chagas disease are considered two of the most relevant neglected diseases. However, the available treatments against these parasites are limited and have multiple disadvantages such as high toxicity, side effects, high costs, the route of administration and the development of resistant strains, among others [5]. In this context, the search for new compounds or chemical models with antiprotozoal activity and low toxicity remains crucial. Moreover, the discovery of an apoptosis-like process in some protozoan parasites such as *Leishmania* and *Trypanosoma* has offered an interesting approach to access new treatments against these parasites. In multicellular organisms, apoptosis is essential for proper development and homeostasis. This mechanism is characterized by different morphological changes that induce cell death without causing inflammation. Some of this apoptosis signals such as cell shrinkage, nuclear chromatin condensation, DNA fragmentation, membrane blebbing, mitochondrial transmembrane potential loss and phosphatidylserine exposure have been found in both parasites, suggesting that controlled cell death of the parasites is possible [6]. In addition, it has been reported that trypanosomatids develop an autophagic genotype as a survival tool under stressful conditions. Autophagy plays an important role in the differentiation of these parasites during their life cycle, the maintenance of homeostasis and the removal of damaged structures. This type of programmed cell death has been reported in *T. cruzi* after the detection of autophagic vacuoles, the presence of myelin-like structures and the involvement of key autophagy protein homologues (ATG homologues) [7].

Among marine organisms, seaweeds are producers of unique secondary metabolites that could interfere with parasite cell pathways; hence, they could be useful in the development of novel therapies [8,9,10,11,12,13,14,15]. The therapeutic potential of marine organisms as a source of new antiparasitic molecular scaffolds has inspired several investigations in our research group [16,17,18]. Secondary metabolites produced by algae include meroterpenoids, which have a wide range of biological activities such as antimicrobial, cytotoxic, antioxidant, anti-inflammatory, antiviral, enzyme inhibitory and immunosuppressive effects [19]. Moreover, the algae belonging to the genus *Gongolaria* spp. contain meroterpenoids, which have reported also anticancer, antimalarial and leishmanicidal activities [11,13,15,20]. Therefore, this type of organism seems to be a promising source of interesting molecules for the development of new antiprotozoal drugs. The aim of the present study was to carry out the bioguided fractionation of an active extract of the brown alga *Gongolaria abies-marina*, testing in vitro its leishmanicidal and trypanocidal activities, as well as its cytotoxicity against murine macrophages. Several chromatographic steps allowed the isolation of four structurally related meroterpenoids, cystomexicone B (**1**), gongolarone B (**2**), 6*Z*-1′-methoxyamentadione (**3**) and 1′-methoxyamentadione (**4**) (Figure 1), which showed antikinetoplastid activity and low cytotoxicity. Activity assays against intracellular amastigotes of *Leishmania amazonensis* of these compounds were performed. Moreover, to investigate whether the antiparasitic effect of the meroterpenoids triggered the induction of programmed cell death of the parasites, different assays to check this were performed.

## 2. Results

### 2.1. Isolation and Chemical Characterization of Meroterpenoids of Gongolaria abies-marina

Specimens of *G. abies-marina* were collected from the intertidal area of the northwest of the island of Tenerife, Spain. After washing and drying, the seaweed was pulverized and macerated at room temperature with dichloromethane and ethyl acetate to obtain a crude extract with leishmanicidal and trypanocidal activities. Specifically, the crude extract showed promising activity against promastigotes of *L. amazonensis* (IC_50_ 5.76 ± 0.06 μg/mL) and epimastigotes of *Trypanosoma cruzi* (IC_50_ 8.37 ± 0.50 μg/mL). However, the extract was not active against promastigotes of *L. donovani* (IC_50_ >100 μg/mL). Based on these data, gel filtration chromatographic fractionation of the crude extract was carried out to afford an active fraction (F6) against all parasites tested, including promastigotes of *L. donovani*. Further fractionation and purification steps allowed the isolation of meroterpenoids cystomexicone B (**1**), gongolarone B (**2**), 6*Z*-1′-methoxyamentadione (**3**) and 1′-methoxyamentadione (**4**). Their structures were confirmed by spectroscopic analysis, and the absolute configuration of the chiral center at C-11 of compounds **2**–**4** was confirmed as 11*R* based on biogenetic considerations and the values of their specific rotations (Figure 1) [16].

### 2.2. Antiparasitic Activity

The activity of the isolated compounds **1**–**4** was determined against *L. amazonensis*, *L. donovani* and *T. cruzi*. The four tested meroterpenoids showed promising antikinetoplastid activity. The compounds with the best leishmanicidal activity in *L. amazonensis* promastigotes were 6*Z*-1′-methoxyamentadione (**3**) and 1′-methoxyamentadione (**4**). These compounds showed similar IC_50_ values around 4.88 µM (Table 1). Furthermore, this activity was slightly higher than the reference drug, miltefosine (Figure 2), with an IC_50_ value of 6.48 µM. Similarly, compounds **3** and **4** showed the best activity results against *L. donovani* with IC_50_ values of 97.94 µM and 126.51 µM, respectively. On the other hand, the compounds with the best trypanocidal activity were gongolarone B (**2**) and 6*Z*-1′-methoxyamentadione (**3**) with IC_50_ values of 2.92 and 3.26 µM, respectively. It should be noted that these IC_50_ values were better than those presented by the reference drug benznidazole (IC_50_ of 6.95 µM) (Figure 2).

The cytotoxicity of compounds **1**–**4** was also tested in murine macrophages. The results showed that cystomexicone B (**1**) was the least toxic compound with a CC_50_ value > 278.96 µM. Based on the selectivity index, cystomexicone B appeared to be the most selective and less toxic for all tested parasites. However, all other isolated meroterpenoids showed selectivity indices 2-fold higher, which was also an indicator of good selectivity towards the parasite [21]. The activity of meroterpenoids **1**–**4** was determined against the intracellular amastigote form of *L. amazonensis* (Table 2). All compounds showed intracellular activity, with IC_50_ values ranging from 16.17 to 27.59 µM. The best activity in amastigotes of *L. amazonensis* and cytotoxicity results were shown by cystomexicone B (**1**), with a selective index of 11.36. Compound **1** was the only compound where the activity in amastigotes of *L. amazonensis* was better than in the promastigote form. However, none of the tested compounds presented better activity than that shown by the reference drug miltefosine.

The next step was to determine the mechanisms of action of the isolated compounds and to define whether these metabolites could induce programmed cell death (apoptosis-like) or if they could cause necrotic death on the parasite.

### 2.3. ATP Level Analysis

The mitochondrial damage was determined by measuring the ATP level after 24 h of treatment using a Cell Titer-Glo^®^ Luminescent Cell Viability Assay (Promega, WI, USA). The analysis of variance was determined by one-way ANOVA and the results are shown in Figure 3. As can be observed, meroterpenoids **1**–**4** significantly decreased the ATP levels produced in the parasites relative to the negative control (untreated parasite). Cystomexicone B (**1**) significantly decreased the ATP levels in all parasites studied, including *L. donovani* (which was the most resistant species of those tested).

In relation to *T. cruzi*, it can be observed that the compounds that most affected the ATP level of the parasite are cystomexicone B (**1**), followed by compounds **2** and **3**. The alterations of ATP production in the parasite caused by compounds **2** and **3** were similar to each other, which was consistent with the activity data shown in Table 1. On the other hand, for *L. amazonensis*, the compound that caused the greatest effect on cellular ATP levels was **3**. Cystomexicone B (**1**) and gongolarone B (**2**) also decreased the levels of ATP on the parasite with no significant differences between them. Although isomers **3** and **4** have similar IC_50_ values (4.88 µM), they affect ATP production differently, while compound **3** reduces it further.

### 2.4. Analysis of Mitochondrial Membrane Potential Changes

For the study of the mitochondrial membrane potential changes, a JC-1 assay was performed. The results of this study are represented in Figure 4. Cystomexicone B (**1**) does not produce significant changes in the mitochondrial membrane potential of any of the parasites tested, despite decreasing ATP levels in all parasites. However, compounds **2**–**4** generally cause significant decreases in mitochondrial membrane potential in *L. amazonensis* and *T. cruzi*. In the case of *T. cruzi*, gongolarone B (**2**) produces the greatest decrease in mitochondrial membrane potential. In addition, compounds **3** and **4** produce similar changes in mitochondrial potential (there are no significant differences between them, unlike what occurred in cellular ATP production). In *L. amazonensis* there are no significant differences between compounds **2**–**4**, similarly decreasing the mitochondrial membrane potential of this parasite.

### 2.5. Plasmatic Membrane Permeability Assay and Chromatin Condensation Analysis

The SYTOX^®^ Green assay was performed to detect alterations of the membrane permeability in treated cells. Appendix A show the results obtained after incubation at the IC_90_ levels of the pure compounds with the three tested parasites: *L. amazonensis* (Appendix A), *L. donovani* (Appendix A) and *T. cruzi* (Appendix A). The images demonstrated that all compounds produced membrane permeability alteration in the parasites as they emitted higher green fluorescence levels than the negative control.

Furthermore, compounds **1**–**4** were shown to induce chromatin condensation as they emitted stronger blue fluorescence levels than the negative control, as can be observed in Appendix A. Some of the parasites emitted a slight red fluorescence, which indicates the presence of dead cells. These results suggest a late apoptotic stage.

### 2.6. Analysis of Reactive Oxygen Species

In order to evaluate the reactive oxygen species levels inside the parasites, CellROX^®^ Deep Red Reagent was used. Appendix A show the results obtained from these experiments after incubation at the IC_90_ of the meroterpenoids with *L. amazonensis* (Appendix A), *L. donovani* (Appendix A) and *T. cruzi* (Appendix A). The images demonstrated that all compounds increase the reactive oxygen species levels inside the parasites, as they emitted higher red fluorescence levels than the negative control.

### 2.7. Immunofluorescence Analysis of Actin

To assess the integrity of the actin network of parasites treated, an immunofluorescence assay was carried out using phalloidin conjugated for actin staining. Appendix A show the results obtained after incubating the parasites at the IC_90_ levels of *L. amazonensis* (Appendix A) and *T. cruzi* (Appendix A). The fluorescence obtained in the images show that in the case of *L. amazonensis*, the actin network seems to be slightly affected by all compounds, since a lower fluorescence level can be observed than in the control. On the other hand, the tested compounds do not appear to damage the actin network of *T. cruzi*. In this case, no differences can be observed between the treated parasites and the control.

### 2.8. Immunofluorescence Analysis of Tubulin

To evaluate the integrity of the tubulin network of parasites, an immunofluorescence assay was carried out using the tubulin antibody as the first antibody and Alexa 594 as the second antibody.

Appendix A show the results obtained from these experiments after incubating the parasites at the IC_90_ levels of *L. amazonensis* (Appendix A) and *T. cruzi* (Appendix A). The fluorescence obtained in the images show that the tubulin network seems to be slightly affected by all tested meroterpenoids in both parasites. The images show how the tubulin network is reduced to small bodies and loses its continuity through the cytoplasm compared to the control. In addition, it highlights how the tubulin present in the flagellum of parasites treated with the compounds of *G. abies-marina* is completely lost.

### 2.9. Ultrastructural Effects on L. amazonensis Promastigotes and T. cruzi Epimastigotes

In order to evaluate the ultrastructural and morphological alterations induced in the parasites by gongolarone B (**2**), 6*Z*-1′-methoxyamentadione (**3**) and 1′-methoxyamentadione (**4**), transmission electron microscopy (TEM) was performed (Figure 5 and Figure 6).

In the *L. amazonensis* control (Figure 5A), the elongated morphology characteristic of promastigotes of this parasite was observed. The cytoplasm and cell organelles were in perfect conditions. After incubating *L. amazonensis* promastigotes at the IC_90_ levels of compounds **2**–**4** (Figure 5B–I), it was confirmed that this concentration of compounds eliminates 90% of the parasites, as it was observed that in most of the parasites the cytoplasm was completely disorganized and empty (black triangles). Overall, the pure compounds caused similar damage to *L. amazonensis* promastigotes. As can be seen in Figure 5B–I, the parasites lost their characteristic elongated morphology without compromising the cell membrane. Furthermore, morphological alterations such as chromatin condensation (*) and swelling of the mitochondria (M) and the kinetoplast (K) were observed.

In addition, a higher number of vacuoles was observed in parasites treated with the meroterpenoids compared to the control. In particular, different types of vacuoles were observed, some of which were well delimited by the vacuolar membrane (V), while in others the surrounding membrane was missing (indicating high intracellular damage) (black star, Figure 5F), with vacuoles with cellular contents and membranoid structures inside, suggesting a process of autophagy (black arrows).

Another notable morphological change was the alteration of the flagellar pocket. This was observed when the parasites were incubated with 6*Z*-1′-methoxyamentadione (**3**) and 1′-methoxyamentadione (**4**) (Figure 5E,F,H). Thus, it was shown that compounds **3** and **4** not only affected essential organelles such as the mitochondrion, but also the motility of the parasite. According to the Figure 5I, the parasites showed two nuclei after incubation with compound **4**, suggesting effects on the cell cycle.

In relation to *T. cruzi*, the negative control (Figure 6A) showed the elongated morphology characteristic of the parasite. The cytoplasm and cell organelles were also observed to be in perfect condition. After incubating *T. cruzi* epimastigotes for 24 h at the IC_90_ levels of gongolarone B (**2**), 6*Z*-1′-methoxyamentadione (**3**) and 1′-methoxyamentadione (**4**), similar ultrastructural damages were observed (Figure 6B–J). The parasites showed alterations and discontinuities in their cytoplasm with respect to the control. The epimastigotes lost their characteristic elongated morphology without compromising the cell membrane. This confirmed that there was no cell lysis, thereby avoiding a secondary inflammatory process. Morphological alterations of the mitochondria and kinetoplast (Figure 6C,E) were detected after incubation of the parasites with gongolarone B (**2**) and 6*Z*-1′-methoxyamentadione (**3**). As is possible to observe in Figure 6B,C,F, the parasites showed condensed chromatin (*) after incubation with gongolarone B (**2**) and 6*Z*-1′-methoxyamentadione (**3**).

Meroterpenoids **2**–**4** induced intense cellular vacuolization (V) in the parasites. Some of the vacuoles presented membranous contents inside, meaning they could be autophagic vacuoles (black arrows). In addition, the increase in reservosomes (R) in the treated parasites of the present work is notable. These organelles are essential for the differentiation of epimastigotes in the gut of the vector. Reservosomes are endocytic compartments that store endocytic macromolecules (such as proteins or lipids) and lysosomal enzymes [22]. Additionally, changes in the Golgi (G) such as an increased number and size of Golgi cisternae followed by increased distension of the endoplasmic reticulum (ER) were observed compared to the control (Figure 6D,G,J).

## 3. Discussion

Herein, the leishmanicidal and trypanocidal activity levels of four meroterpenoids isolated from *G. abies-marina*, cystomexicone B (**1**), gongolarone B (**2**), 6*Z*-1′-methoxyamentadione (**3**) and 1′-methoxyamentadione (**4**), were analyzed. These compounds showed IC_50_ values ranging from 4.88 to 36.65 µM for promastigotes of *L. amazonensis* and from 2.92 to 19.19 µM for epimastigotes of *T. cruzi*. These activity values were very similar to those of the reference drugs miltefosine and benznidazol (IC_50_ 6.48 µM, and 6.95 µM, respectively). The activity of meroditerpenoids against *L. amazonensis* amastigotes was also calculated, with IC_50_ values ranging from 16.17 to 27.59 µM. These results are in consonance with the literature consulted.

Previous studies reported the leishmanicidal activity levels of meroterpenoids from algae belonging to the genus *Gongolaria* spp., previously known as *Cystoseira* spp. Bruno de Sousa et al. reported the activity against promastigotes of *Leishmania infantum* of the meroditerpenoids obtained from *Cystoseira baccata*, 3R-tetraprenyltoluquinone and 3S-tetraprenyltoluquinone, with IC_50_ values of 44.90 and 94.40 µM, respectively. These isomers were also shown to be active against the intracellular form of *L. infantum,* with IC_50_ levels of 25 and 88 µM, respectively [13]. In addition, the leishmanicidal activity levels of extracts of different species of algae belonging to the genus *Cystoseira* spp. (*C. baccata*, *Cystoseira barbata*, *Cystoseira tamariscifolia* and *Cystoseira usneoides*) against both promastigotes and amastigotes of *L. infantum* have been reported [23], revealing the biological interest in the molecules biosynthesized within this genus. Furthermore, some crude extracts from *C. baccata* and *C. tamariscifolia* have demonstrated activity against the promastigotes’ *L. donovani* and trypanocidal activity (determined against *T. cruzi* and *Trypanosoma brucei rhodoniense*) [15], consolidating the results obtained in the present work and the promising leishmanicidal and trypanocidal activity levels of the extracts and compounds isolated from algae of the genus *Gongolaria* spp.

An analysis of the ultrastructural effects of compounds isolated from *G. abies-marina* on *L. amazonensis* and *T. cruzi* revealed the presence of autophagy vacuoles after incubation with meroterpenoids **2**–**4**. The presence of autophagic vacuoles in *Leishmania* spp. is a common feature found in the effects of meroditerpenoids isolated from this alga [24]. Bruno de Sousa et al. isolated from *C. baccata* two meroterpenoids that induced the formation of multilamellar structures that could be compatible with autophagic processes, as observed in the present study [13]. Gonzalez et al. reported high vacuolization levels similar to those observed in this study when incubating *L. infantum* promastigotes with C-20 diterpenoid alkaloid derivatives [24]. Moreover, autophagy vacuoles have also been observed after incubation with an inhibitor of subtilisin (important protease of *Leishmania* spp.) in *L. infantum* [25]. In relation to *T. cruzi,* in the literature other studies also report the formation of autophagy vacuoles after the incubation of parasites with different types of inhibitors such as natural sesquiterpenoids [26] or naphthoquinones derivatives [27]. In the latter case, Lara et al. obtained evidence that naphthoquinone derivatives can trigger the process of type II programmed cell death (autophagic cell death) and was morphologically characterized by the observation of vacuoles containing membrane structures (autophagosomes) such as those reported in the present study. Therefore, the results obtained and evidence of autophagy in these parasites prove that meroterpenoids isolated from *G. abies-marina* could induce an autophagy process in *L. amazonensis* and *T. cruzi*.

The different assays performed to elucidate the physiological changes induced by meroterpenoids in the treated parasites suggest that meroterpenoids **1**–**4** induce apoptosis-like effects in *L. amazonensis*, *L. donovani* and *T. cruzi,* as has been previously reported in other studies [7]. For instance, meroterpenoids **1**–**4** produce changes in parasite membrane permeability without compromising parasite cell integrity. This can be observed in assays performed using the SYTOX^®^ Green dye as well as in transmission electron microscopy images (TEM). A characteristic sign of apoptosis is chromatin condensation could be detected after the incubation of *L. amazonensis*, *L. donovani* and *T. cruzi* with the isolated meroterpenoids. Chromatin condensation was detected both with the Hoechst 33,342 kit and TEM. Compounds isolated from *G. abies-marina* also induced intracellular ROS accumulation in parasites. Alterations in mitochondria and kinetoplast morphology occurred. TEM imaging showed that meroterpenoids **2**–**4** caused alterations in the morphologies of these two organelles in *L. amazonensis* and *T. cruzi*. These mitochondrial alterations were also manifested by decreased cellular ATP levels and lower mitochondrial membrane potential. These changes induced by the meroterpenoids in the present research have been previously reported in kinetoplastids and are associated with an apoptosis-like process. For instance, Chiboub et al. reported that three diterpenes isolated from another brown seaweed (*Dictyota spiralis*) induced apoptotic-like cell death in *L. amazonensis* and *T. cruzi*. These diterpenes also triggered the collapse of the mitochondrial membrane potential, decreased ATP levels, chromatin condensation, the accumulation of ROS, maintenance of the membrane permeability and changes in the cell’s morphology, as reported in the present study [28]. Moreover, the mitochondrial swelling and kinetoplast morphological alterations observed in the present study have been reported previously in *Leishmania* spp. and *T. cruzi*. Gonzalez et al. reported that C-20 diterpenoid alkaloid derivatives induced mitochondrial swelling, vacuolization and kinetoplast disorganization in *L. infantum* [22].

Actin is a major constituent of the cytoskeletons of many organisms, including these protozoa. Actin and actin-binding proteins of kinetoplastids have been shown to perform diverse functions such as regulating endocytosis and intracellular trafficking. Actin is also involved in the dynamics and assembly of the *Leishmania* flagellum, as well as coordinating karyokinesis by effecting spindle elongation and DNA synthesis [29]. Beside inducing programmed cell death via autophagy and apoptosis-like effects, assays performed in this study showed that meroterpenoids **2**–**4** could target different cellular organelles in *L. amazonensis* and *T. cruzi*. For instance, in *L. amazonensis,* these meroterpenoids affect the actin distribution network, whereas in *T. cruzi* this element of the cytoskeleton is not affected. Based on the results obtained in the present study, these actin functions could be affected in *L. amazonensis* by meroterpenoids isolated from *G. abies-marina*. Furthermore, according to TEM images in the present study, 1′-methoxyamentadione (**4**) could arrest the parasite cell cycle in the G2/M phase of *L. amazonensis,* as previously has been reported by Angelo do Souza et al. after the incubation of histone deacetylases inhibitors in *Leishmania brasilienzis* [30].

The tubular network plays an important role in motility and differentiation during the trypanosomatid life cycle [31]. Another important target appears to be the parasite tubulin network, which was strongly affected by compounds **2**–**4** in *L. amazonensis* and *T. cruzi,* altering the motility and differentiation of the tested parasites. This evidence was also reinforced by TEM imaging, as it was observed that meroterpenoids **3** and **4** induced alterations in the flagellar pocket of *L. amazonensis*, consistent with the ultrastructural damage. According to the literature, Bruno de Sousa et al. also reported alterations in the flagellar morphology and motility in promastigotes of *L. infantum* after incubation with crude extracts (containing meroterpenoids) from seaweeds of the same genus as *Gongolaria* (*C. baccata* and *C. barbata*) [23].

After incubation with meroterpenoids **2**–**4**, an increase in reservosomes associated with changes in the Golgi complex and distension of the ER in *T. cruzi*. was observed. Engel et al. also reported Golgi enlargement followed by ER distension after incubating *T. cruzi* epimastigotes with cysteine protease inhibitors. These results suggested that cruzain accumulation may decrease the mobility of Golgi membranes and cause the peripheral distension of the cisternae [32]. In addition, another study by Salomao et al. reported ultrastructural alterations of reservosomes, the Golgi complex and the ER similar to those reported after incubating *T. cruzi* epimastigotes with Brazilian green propolis ethanolic extract. According to Salmao et al., the increased amounts of reservosomes and the lipids stored inside them, together with the morphological and structural changes of the Golgi and ER, suggest an impairment of the endocytic pathway of the parasite leading to cell death. This could explain the ultrastructural alterations observed in *T. cruzi* when incubated with compounds **2**–**4**. This evidence could indicate that the alterations in the Golgi complex induced by **2**–**4** could compromise the glycosylation or transport of proteins from the endoplasmic reticulum to the secretory vacuoles and lysosomes [22]. Thus, meroterpenoids **2**–**4** seem to target reservosomes, the Golgi complex and the ER of *T. cruzi*. However, further studies with meroterpenoids **2**–**4** should be performed to elucidate how reservosome formation is triggered in the case of these compounds isolated from *G. abies-marina*.

Under the chemical point of view, compounds **2**–**4** are structural isomers, whereas cystomexicone B (**1**) can be considered a *nor*-C_6_-metabolite obtained from the oxidative fragmentation of 1′-methoxyamentadione (**4**). Chemical differences that justify the observed antiparasitic activity seem to be related to the positional and geometric isomerization of double bonds in fragment C6–C8 (Figure 7). Thus, the *E*/*Z* geometric isomerization of double bonds between C6–C7 do not affect the activity of isomers **3** and **4** against promastigotes of *Leishmania* species. However, 6*Z*-1′-methoxyamentadione (**3**) showed better activity against *T. cruzi*. On the other hand, the positional isomerization of *E*-C6-C7 (**4**) into *E*-C7–C8 (**2**) does not significantly affect the activity against *L. amazonensis*. Thus, gongolarone B (**2**) showed selective activity and revealed good IC_50_ values against *L. amazonensis,* while it lost its activity against *L. donovani* and remained the most active against *T. cruzi.* Finally, the oxidative fragmentation at C11-C12 to afford the *nor*-C_6_-meroditerpene, cystomexicone B (**1**), drastically reduced the antiparasitic activity and the toxicity.

According to the evidence found in this study and the literature, meroterpenoids **2**–**4** could initially induce an autophagy process in kinetoplastids as a survival mechanism. If the cellular stress conditions induced by the compounds in the parasites continue, apoptosis-like is then activated. This hypothesis is consistent with previous research linking apoptosis and autophagy. Different examples of autophagy and apoptosis-like processes after the incubation of kinetoplastids with different compounds were reported in a recent review by Das P. et al. [33]. For instance, cryptolepine generates autophagic vacuoles in *L. donovani,* and from them different events characteristic of apoptosis such as DNA fragmentation occur. In addition, the present meroterpenoids may have other targets in parasites that may be the key to their good antiparasitic activity. Thus, meroterpenoids affect the tubulin network and the motility of the flagellum of kinetoplastids and the actin network of *L. amazonensis*, so endocytosis, intracellular trafficking, karyokinesis and the assembly of flagellum could be altered [23,30]. Furthermore, changes in the endocytic pathway of the parasites via alterations in the ER and Golgi complex have been observed. Likewise, alterations in the Golgi complex induced by these compounds could compromise glycosylation or protein transport from the endoplasmic reticulum to the secretory vacuoles and lysosomes [22]. This makes this family of compounds potential scaffolds for drug design against leishmaniasis and Chagas disease.

## 4. Materials and Methods

### 4.1. General Chemical Methods and Procedures

NMR spectra were acquired on a Bruker AVANCE 500 MHz or 600 MHz (Bruker Biospin, Falländen, Switzerland) instrument spectrometer at 300 K when required. The Bruker AVANCE 600 MHz spectrometer was equipped with a 5 mm TCI inverse detection cryoprobe (Bruker Biospin, Falländen, Switzerland). Standard Bruker NMR pulse sequences were utilized. NMR spectra were obtained by dissolving samples in CDCl_3_ (99.9%). Optical rotations were measured in CHCl_3_ on a PerkinElmer 241 polarimeter (Waltham, MA, USA) by using a Na lamp. IR spectra were recorded using an Agilent Cary 630 FTIR spectrometer (Agilent Technologies, Inc., Santa Clara, CA, USA) equipped with an ATR unit. HPLC (high-performance liquid chromatography) separations were carried out with an Agilent 1260 Infinity Quaternary LC system equipped with a diode array detector (Waldbronn, Germany). Thin-layer chromatography (TLC) silica gel plates were used to monitor the column chromatography, visualized by UV light (254 nm) and developed with cobalt chloride (2%) as a spraying reagent. All reagents and solvents were commercially available and used as received.

### 4.2. Algae Material

Algae specimens [16] were collected off the intertidal zone of the coast of Bajamar, Tenerife, Canary Islands (28°33′15.5″ N 16°20′51.7″ W), in April 2019; transported to the laboratory; and cleaned, rinsed and dried in the dark. The algae were identified as *Gongolaria abies-marina* by Dr. M. Sansón (Department of Marine Botany of Universidad de La Laguna). A voucher specimen is deposited at the Herbario TFC of SEGAI-ULL under the code 11042019-3.

### 4.3. Extraction and Bio-Guided Fractionation: Isolation of Compounds ***1***–***4***

The dried algal material (233.4 g) was ground and sequentially macerated at room temperature in dichloromethane (DCM) and ethyl acetate (EtOAc) for extraction. Each solvent was renewed three times. The organic solvents were combined, filtered, and evaporated to obtain the crude extract (1.94 g). The extract was fractionated by gel filtration chromatography (Sephadex LH-20 column, *n*-hexane/DCM/methanol (7:2:1)) to afford fractions F1–F6. Size-exclusion chromatography in Sephadex LH-20 (*n*-hexane/DCM/methanol (3:1:1)) of the active fraction F6 (689 mg), followed by medium-pressure chromatography of fraction F6.4 (257.4 mg) with a Lobar LiChroprep Si 60 (40–63 μm) column (*n*-hexane/EtOAc (1:1)) and silica gel column (step gradient from CHCl_3_/EtOAc (4:1) to 100% EtOAc) of fraction F6.4.3 (110 mg), allowed the isolation of cystomexicone B (**1**, 1.39 mg). The normal-phase HPLC of fraction F6.4.3-5 (51.86 mg) (Phenomenex, Luna 5 μm Silica (2) column, 100 Å, 250 × 10 mm; isocratic *n*-hexane/EtOAc (3:2), 10 min at 1 mL/min; gradient up to 100% EtOAc, 50 min at 2 mL/min; 100% EtOAc, 5 min, 2 mL/min) afforded the pure compounds gongolarone B (**2**, 1.75 mg, 34.1 min), 6*Z*-1′-methoxyamentadione (**3**, 3.10 mg, 37.1 min) and 1′-methoxyamentadione (**4**, 6.22 mg, 42.1 min). The physical properties and NMR data of compounds **1**–**4** were confirmed based on those previously reported [16,34].

#### 4.3.1. Cystomexicone B (**1**)

Colorless oil; IR ν_max_ 3600, 3350, 1750, 1675, 1450, 1358, 1310, 1160 cm^−1^; HRESIMS *m*/*z* 381.2040 [M+Na]^+^ (calc. C_22_H_30_O_4_Na, 381.2036); ^1^H NMR (600 MHz, CDCl_3_) δ 6.53 (s, 2H), 6.18 (t, *J* = 1.4 Hz, 1H), 6.01 (s, 1H), 5.47–5.42 (m, 1H), 3.68 (d, *J* = 0.9 Hz, 3H), 3.40 (d, *J* = 7.1 Hz, 2H), 3.10 (s, 2H), 2.46 (t, *J* = 7.0 Hz, 2H), 1.80–1.74 (m, 3H). ^13^C NMR (150 MHz, CDCl_3_) δ 210.1, 199.7, 158.3, 152.3, 150.3, 134.8, 132.3, 131.1, 128.3, 122.4, 115.9, 113.74, 60.7, 56.1, 42.5, 40.1, 30.3, 28.1, 21.5, 19.4, 16.5, 16.4.

#### 4.3.2. Gongolarone B (**2**)

Colorless oil; [α]_D_^20^ -22 (*c* 0.17, CHCl_3_); IR ν_max_ 2970, 2929, 1699 cm^−1^; HRESIMS *m*/*z* 479.2770 [M+Na]^+^ (calc. C_28_H_40_O_5_Na, 479.2773); ^1^H (600 MHz, CDCl_3_) δ 6.94 (d, *J* = 15.6 Hz, 1H), 6.57 (d, *J* = 3.1 Hz, 1H), 6.54 (d, *J* = 3.1 Hz, 1H), 6.39 (d, *J* = 15.6 Hz, 1H), 5.37 (ddd, *J* = 7.5, 7.5, 1.3 Hz, 1H), 5.30 (ddd, *J* = 7.2, 7.2, 0.9 Hz, 1H), 3.67 (s, 3H), 3.38 (d, *J* = 7.5 Hz, 2H), 3.20 (s, 2H), 3.13 (d, *J* = 5.0 Hz, 1H), 2.76 (ddd, *J* = 8.7, 6.8, 4.9 Hz, 1H), 2.25 (s, 3H), 1.93 (dddd, *J* = 13.9, 7.2, 7.2, 7.2 Hz, 1H), 1.71 (d, *J* = 0.9 Hz, 3H), 1.71 (m, 1H), 1.68 (d, *J* = 1.3 Hz, 3H), 1.51 (m, 1H), 1.40 (s, 6H), 1.08 (d, *J* = 6.8 Hz, 3H); ^13^C NMR (150 MHz, CDCl_3_) δ 208.7, 204.9, 152.8, 152.6, 149.7, 133.6, 132.2, 130.5, 129.2, 129.0, 128.1, 124.5, 115.8, 113.3, 71.7, 60.5, 53.4, 46.3 44.5, 33.6 29.1, 29.1, 27.5, 26.2, 24.6, 17.0, 16.7, 16.3

#### 4.3.3. 6*Z*-1′-Methoxyamentadione (**3**)

Colorless oil; [α]_D_^20^ -34 (*c* 0.06, CHCl_3_); IR ν_max_ 3600, 3420, 2905, 1703, 1685, 1435, 1358 cm^−1^; HRESIMS *m*/*z* 479.2775 [M+Na]^+^ (calc. C_28_H_40_O_5_Na, 479.2773); ^1^H NMR (600 MHz, CDCl_3_) δ 6.97 (d, *J* = 15.7 Hz, 1H), 6.86 (d, *J* = 1.3 Hz, 1H), 6.58 (d, *J* = 3.2 Hz, 1H), 6.56 (d, *J* = 15.7 Hz, 1H), 6.53 (d, *J* = 3.0 Hz, 1H), 6.08 (s, 1H), 5.35 (t, *J* = 7.7 Hz, 1H), 3.67 (s, 3H), 3.36 (d, *J* = 7.6 Hz, 2H), 3.14 (s, 2H), 2.89–2.72 (m, 2H), 2.58 (td, *J* = 10.8, 5.9 Hz, 1H), 2.47 (td, *J* = 10.8, 5.7 Hz, 1H), 2.25 (s, 3H), 1.85 (d, *J* = 1.4 Hz, 3H), 1.75 (dtd, *J* = 17.1, 8.4, 2.9 Hz, 1H), 1.69 (s, 2H), 1.54 (ddd, *J* = 15.5, 7.8, 5.9 Hz, 1H), 1.38 (s, 5H), 1.10 (d, *J* = 6.9 Hz, 3H).^13^C NMR (151 MHz, CDCl_3_) δ 205.1, 199.9, 152.9, 152.6, 149.9, 133.9, 132.1, 131.2, 127.6, 124.3, 123.9, 115.6, 113.7, 71.6, 60.6, 55.1, 44.5, 34.1, 33.9, 29.2, 29.1, 27.6, 25.6, 17.1, 16.8, 16.4.

#### 4.3.4. 1′-Methoxyamentadione (**4**)

Colorless oil; [α]_D_^20^ -13 (*c* 0.09, CHCl_3_); IR ν_max_ 3700, 3435, 2900, 1680, 1455, 1355 cm^−1^; HRESIMS *m*/*z* 455.2796 [M+H]^+^ (calc. C_28_H_39_O_5_, 455.2797); ^1^H NMR (600 MHz, CDCl_3_) δ 6.94 (d, *J* = 15.6 Hz, 1H), 6.55 (d, *J* = 3.1 Hz, 0H), 6.53 (d, *J* = 3.1 Hz, 0H), 6.39 (d, *J* = 15.6 Hz, 1H), 6.34 (s, 0H), 6.12 (s, 1H), 5.42 (t, *J* = 7.4 Hz, 1H), 3.68 (s, 2H), 3.38 (d, *J* = 7.3 Hz, 1H), 3.12 (s, 1H), 2.73 (h, *J* = 6.8 Hz, 1H), 2.25 (s, 2H), 2.11 (p, *J* = 7.4 Hz, 1H), 2.06 (s, 2H), 1.70 (s, 2H), 1.47–1.41 (m, 1H), 1.39 (d, *J* = 2.3 Hz, 4H), 1.10 (d, *J* = 6.9 Hz, 2H). ^13^C NMR (151 MHz, CDCl_3_) δ 204.3, 200.1, 158.6, 152.9, 152.2, 149.9, 134.5, 132.0, 131.0, 127.8, 123.7, 122.8, 115.5, 113.7, 71.3, 60.5, 55.4, 44.8, 40.9, 32.1, 29.3, 29.3, 27.9, 24.5, 19.4, 16.5, 16.4, 16.2.

### 4.4. Cultures

To carry out the experiments, promastigotes of *Leishmania amazonensis* (MHOM/BR/77/LTB0016) and *L. donovani* (MHOM/IN/90/GE1F8R) and epimastigotes of *Trypanosoma cruzi* (Y strain) were employed. *Leishmania* species were cultured in Schneider’s medium (Sigma-Aldrich, Madrid, Spain), supplemented with 10% fetal bovine serum (VWR, Biowest, Nuaillé, France) and incubated at 26 °C. RPMI 1640 medium (Gibco, Thermo Fisher, Madrid, Spain) with or without phenol red was also used to culture the parasites at the same temperature. Epimastigotes of *T. cruzi* were cultured in liver infusion tryptose (LIT) medium supplemented with 10% fetal bovine serum at 26 °C. A cytotoxicity assay was performed using a murine macrophage cell line J774A.1 (ATCC TIB-67), which was maintained in Dulbecco’s modified Eagle’s medium (Gibco, Thermo Fisher, Madrid, Spain) supplemented with 10% of fetal bovine serum at 37 °C in a 5% CO_2_ atmosphere.

### 4.5. Leishmanicidal and Trypanocidal Activity Assay

With the objective of determining the antiparasitic activity of the compounds isolated from *G. abies-marina,* leishmanicidal activity assays were performed against promastigote stages of *L. amazonensis* and *L. donovani*, as well as a trypanocidal activity assay performed against epimastigote stages of *T. cruzi.* In both parasites, a colorimetric assay was used based on the Alamar Blue reagent (Life Technologies, Madrid, Spain) as previously described by Bethencourt-Estrella et al. (2021) [35].

In a sterile 96 well-plate, serial dilutions of the extracts, fractions or compounds isolated from *G. abies-marina* were performed in RPMI 1640 medium without phenol red (in the case of *Leishmania* spp.) or in LIT (in the case of *T. cruzi*) with a final volume of 100 µL. Parasites were added to wells to reach a concentration of 5 × 10^5^ (*Leishmania* spp.) or 2.5 × 10^5^ (*T. cruzi*) until a final volume of 200 µL/well. Next, 10% of Alamar Blue^®^ was placed into each well and the plate was incubated for 72 h at 26 °C. Finally, the fluorescence of each well was measured using an EnSpire^®^ Multimode Plate Reader (Perkin Elmer, Madrid, Spain) (544 nm excitation, 590 nm emission) and the concentration that inhibits 50% of the parasite population (IC_50_) was calculated using a non-linear regression analysis with 95% confidence limits. Miltefosine and benznidazol were used as reference drugs. To analyze the data and calculate the IC_50_, SigmaPlot 12.0 statistical analysis software (Systat Software Inc, Palo Alto, CA, USA) was used.

### 4.6. In Vitro Effect against Amastigote Stage of Leishmania amazonensis

To determine the in vitro effect against the amastigote stage of *L. amazonensis,* murine macrophages were infected. In a sterile 96 well-plate, macrophages of the J774A.1 cell line were placed in RPMI 1640 to reach 10^5^ macrophages per well. This plate was incubated at 37 °C in a 5% CO_2_ atmosphere. After one hour of incubation, 100 µL of stationary-phase promastigotes of 7-day-old culture was added in a 10:1 ratio (10^6^ parasites per well) and the plates were re-incubated at 37 °C for 24 h to achieve maximum infection. Then, the wells were washed with fresh medium to remove non-phagocytosed promastigotes. The infected macrophages were treated with the compounds isolated for 24 h. Then, the medium was removed carefully to be replaced by 30 μL of 0.05% SDS and the plate was shaken for 30 s. After broke macrophages, 170 μL of Schneider’s medium was added to each well to give a final volume of 200 μL and 20 μL of Alamar Blue^®^ was added to the plate being incubated at 26 °C for 72 h, as described by López-Arencibia et al. (2017) [36]. After 72 h of incubation, the plates were analyzed using the same protocol for the promastigote test.

### 4.7. Cytotoxicity Assay

In a sterile 96-well plate, murine macrophages of the cell line J774A.1 were placed to reach an amount of 10^5^ cells per well using RPMI 1640 medium without phenol red. After murine macrophage attachment, 50 µL serial dilutions of the isolated compounds were added in each well until reaching a final volume of 100 µL. Next, 10% of Alamar Blue^®^ was placed into each well and the plate was incubated for 24 h at 37 °C with a 5% CO_2_ atmosphere. Finally, the plates were analyzed using an EnSpire^®^ Multimode Plate Reader (PerkinElmer, Madrid, Spain), as described in the previous section, and the 50% cytotoxic concentration (CC_50_) values were calculated.

### 4.8. Study of the Mechanism of Cell Death in the Parasites

To study the mechanism of cell death induced in parasites by compounds isolated from *Gongolaria abies-marina*, a common step was carried out to perform the following five experiments. The promastigotes of *Leishmania* spp. and epimastigotes of *T. cruzi* were incubated at the IC_90_ levels of the compounds (previously calculated) for 24 h at 26 °C. After incubation, the parasites were centrifuged at 3000 rpm for 10 min at 4 °C. The pellets were resuspended following the manufacturer’s instructions for each kit.

#### 4.8.1. Analysis of ATP Levels

After the incubation of promastigotes and epimastigotes of the parasites with the IC_90_ of the compounds, the changes in ATP levels were measured using a Cell Titer-Glo^®^ Luminescent Cell Viability Assay (Promega, WI, USA). The assay was carried out in a white 96-well plate following the manufacturer’s instructions, as previously described by Cartuche et al. (2020) [37]. The luminescence was measured using an EnSpire^®^ Multimode Plate Reader (PerkinElmer, Madrid, Spain).

#### 4.8.2. Analysis of Mitochondrial Membrane Potential

A JC-1 Mitochondrial Membrane Potential Assay Kit^®^ (Cayman Chemical, Ann Arbor, MI, USA) was used to analyze variations in the mitochondrial membrane potential of the parasites after incubation at the IC_90_ levels of the compounds. After centrifugation, the parasites were resuspended in the same buffer and transferred to a black 96-well plate. JC-1 was added following the manufacturer’s instructions as also described by López-Arencibia et al. (2020) [38]. Green and red fluorescence was measured using an EnSpire^®^ Multimode Plate Reader (PerkinElmer, Madrid, Spain).

#### 4.8.3. Analysis of Plasma Membrane Permeability

To study the variations in plasma membrane permeability of the parasites after incubation with the IC_90_ levels of the compounds, SYTOX^®^ Green nucleic acid stain fluorescent dye (ThermoFisher Scientific, Waltham, MA, USA) was used. The assay was performed following the manufacturer’s instructions and the fluorescence was monitored by taking pictures after 15 min of incubation with the EVOS^®^ FL Cell Imaging System (ThermoFisher Scientific, Waltham, MA, USA) [39].

#### 4.8.4. Chromatin Condensation Determination

One of the characteristic steps in apoptosis-like cell death is the chromatin condensation in the nucleus of the cells. To determine the presence of this event, Vybrant^®^ Apoptosis Assay Kit n°5, Hoechst 33342/Propidium Iodide (ThermoFisher Scientific, Waltham, MA, USA), was used as recommended by the manufacturer. After incubation and centrifugation, the parasites were incubated with Hoechst 33,342 at 5 μg/mL and PI at 1 μg/mL as previously described [40]. After 15 min of incubation at 26 °C, the fluorescence was monitored by taking pictures with the EVOS^®^ FL Cell Imaging System (ThermoFisher Scientific, Waltham, MA, USA).

#### 4.8.5. Analysis of Reactive Oxygen Species

To measure the accumulation of reactive oxygen species in the parasites incubated at the IC_90_ levels of the isolated compounds, CellROX^®^ Deep Red Reagent (ThermoFisher Scientific, Waltham, MA, USA) was used following the manufacturer’s instruction. The parasites were incubated with 5 μM of CellROX reagent for 30 min at 26 °C. A positive control with H_2_O_2_ at 600 μM was used. Finally, the fluorescence was analyzed by taking pictures with the EVOS^®^ FL Cell Imaging System (ThermoFisher Scientific, Waltham, MA, USA) [41].

#### 4.8.6. Immunofluorescence Analysis of Actin

After 24 h incubation of promastigotes of *L. amazonensis* and epimastigotes of *T. cruzi* at the IC_90_ levels of the pure compounds, the cells were centrifuged at 1500 rpm at 4 °C for 10 min. Next, 50 µL of the pellet was deposited on a pre-coated coverslip and rested for 1 h. The parasites were fixed with formaldehyde (4%) for 30 min, washed in PBS 1X and treated with Triton (0.3%) for 10 min. For direct staining fluorescence, the cells were incubated with phalloidin–tetramethylrhodamine B isothiocyanate (phalloidin-TRITC; SigmaAldrich, Madrid, Spain) for 1 h at room temperature. Finally, the cells were washed with PBS 1X and a drop of mounting DAPI solution was added (4′,6-Diamidino-2-phenylindole dihydrochloride; Sigma-Aldrich; Madrid). The promastigotes and epimastigotes of the parasites tested were examined by Z-stack imaging using an EVOS™ FL Cell Imaging System M5000 (Life Technologies, EE. UU.) at λ_exc_ = 540 nm and λ_em_ = 570 nm.

#### 4.8.7. Immunofluorescence Analysis of Tubulin

To study the intracellular tubulin distribution, promastigotes of *L. amazonensis* and epimastigotes of *T. cruzi* were incubated with the IC_90_ of the pure compounds for 24 h. Next, treated parasites and negative controls were centrifuged at 1500 rpm at 4 °C for 10 min. Then, 50 µL of the pellet was deposited on a pre-coated coverslip and rested for 1 h. The parasites were fixed with formaldehyde (4%) for 15 min, washed in PBS 1X and treated with Triton (0.3%) for 10 min. The parasites were treated with 5% BSA in PBS 1X/150 mM saccharose for 30 min and washed with glycine 100 mM in PBS 1X for 5 min. For incubation of the cells with the primary antibody, the tubulin antibody was added at 1:2000 (monoclonal anti-α-tubulin antibody produced in mice; Sigma-Aldrich, Madrid) and it was incubated for 2 h at room temperature. Alexa 594 1:500 (goat anti-mouse IgG (H+L) Highly Crossed Adsorbed Secondary Antibody, Alexa Fluor Plus 594; Thermo Fisher Scientific, Rockford, IL, USA) was used as the secondary antibody and incubated with the cells for one hour at room temperature in the dark. Finally, the cells were washed with PBS 1X and a drop of mounting DAPI solution was added (4′,6-diamidino-2-phenylindole dihydrochloride; Sigma-Aldrich; Madrid). Z-stack imaging using an inverted confocal microscope Leica DMI 4000 B with a 63× objective (Leica Microsystems, Wetzlar, Germany) was used to examine the promastigotes and epimastigotes of the tested parasites.

### 4.9. Transmission Electron Microscopy (TEM)

The analysis of the ultrastructural and morphological damage caused by the compounds in the parasites was performed in collaboration with the Facultad de Estudios Superiores Iztacala (FESI), Medicina, UNAM (Mexico). These assays were performed after incubation of the promastigotes of *L. amazonensis* and epimastigotes of *T. cruzi* at the IC_90_ levels of the pure compounds over 24 h. TEM images were obtained using the conventional microscopy technique as previously described by Castelan-Ramirez et al. [42,43] in a JEOL JEM-1011 transmission electron microscope (JEOL Ltd., Tokyo, Japan).

### 4.10. Statistical Analysis

The 50% inhibitory concentration (IC_50_) and 50% cytotoxic concentration (CC_50_) values were obtained using Sigma Plot 12.0 statistical analysis software (Systat Software) and calculated using a non-linear regression analysis with 95% confidence limits. Leishmanicidal and trypanocidal activity assays, as well as cytotoxicity assays, were performed three times on different days, and the results were expressed as means ± standard deviations. The analysis of variance was determined using a one-way ANOVA using the GraphPad.PRISM^®^ 9.0 software program (GraphPad Software, San Diego, CA, USA). Differences at *p* < 0.05 were considered statistically significant.

## 5. Conclusions

Meroterpenoids **1**–**4** isolated from *G. abies-marina* showed leishmanicidal and trypanocidal activities. These compounds exhibited similar activities to the reference drugs, miltefosine and benznidazole, against promastigotes and amastigotes of *L. amazonensis*, promastigotes of *L. donovani* and epimastigotes of *T. cruzi*. 6*Z*-1′-Methoxyamentadione (**3**) and 1′-methoxyamentadione (**4**) showed leishmanicidal activity against promastigotes of *L. amazonensis,* with IC_50_ values of 4.88 µM in both cases. Gongolarone B (**2**) and 6*Z*-1′-methoxyamentadione (**3**) demonstrated their trypanocidal potential, with IC_50_ values of 2.92 and 3.26 µM, respectively. In general, the *nor*-C_6_ meroditerpene, cystomexicone B (**1**), demonstrated low cytotoxicity. Furthermore, these meroterpenoids induce some changes in the parasites that compromise their survival, such as reductions in ATP levels and mitochondrial membrane potential, the accumulation of reactive oxygen species, chromatin condensation, alterations of the tubulin network, mitochondrial swelling and changes in the morphology of the ER and Golgi complex, as well as the formation of autophagy vacuoles. Therefore, these meroterpenoids could induce the programmed cell death of kinetoplastids via autophagy and apoptosis-like processes. In addition, these compounds may have other targets in parasites that could be the key to their good antiparasitic activity, affecting the tubulin network and the motility of the flagellum of the kinetoplastids. In consequence, this could lead to alterations in the endocytic pathway of the parasites via alterations in the ER and Golgi complex and could compromise the glycosylation or protein transport from the endoplasmic reticulum to the secretory vacuoles and lysosomes.

## Figures and Tables

**Figure 1 pharmaceuticals-16-00476-f001:**
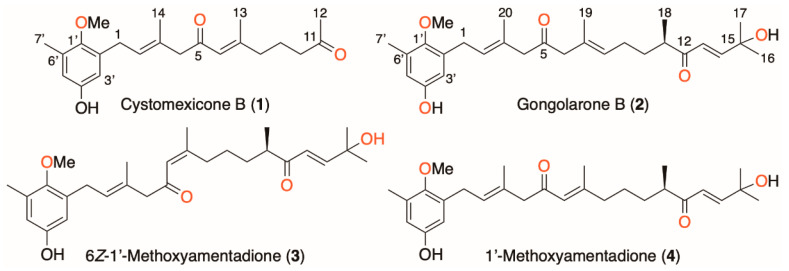
Chemical structure of meroterpenoids isolated from *Gongolaria abies-marina*.

**Figure 2 pharmaceuticals-16-00476-f002:**
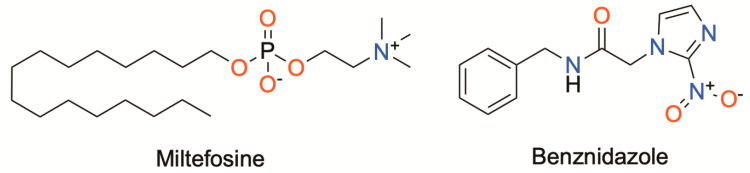
Chemical structures of the reference drugs, miltefosine and benznidazole.

**Figure 3 pharmaceuticals-16-00476-f003:**
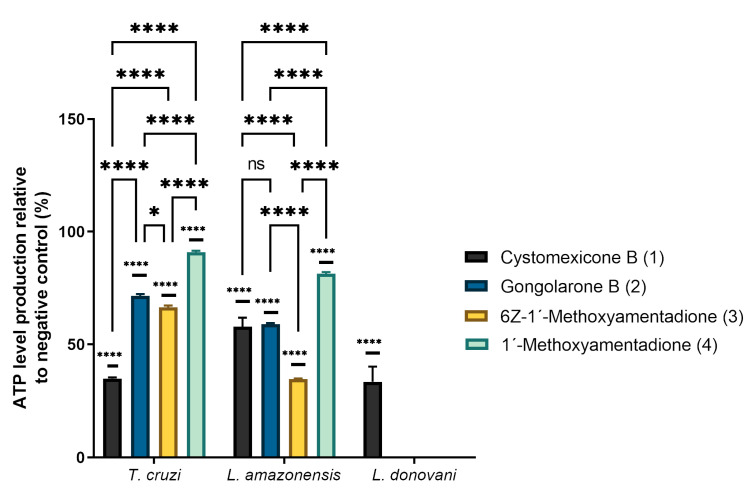
Percentages of ATP levels relative to negative control. The analysis of variance was determined by one-way ANOVA using GraphPad.PRISM^®^ 9.0 software. Significance differences when comparing different percentages values are represented as ns = non-significant; * *p* < 0.1 and **** *p* < 0.0001.

**Figure 4 pharmaceuticals-16-00476-f004:**
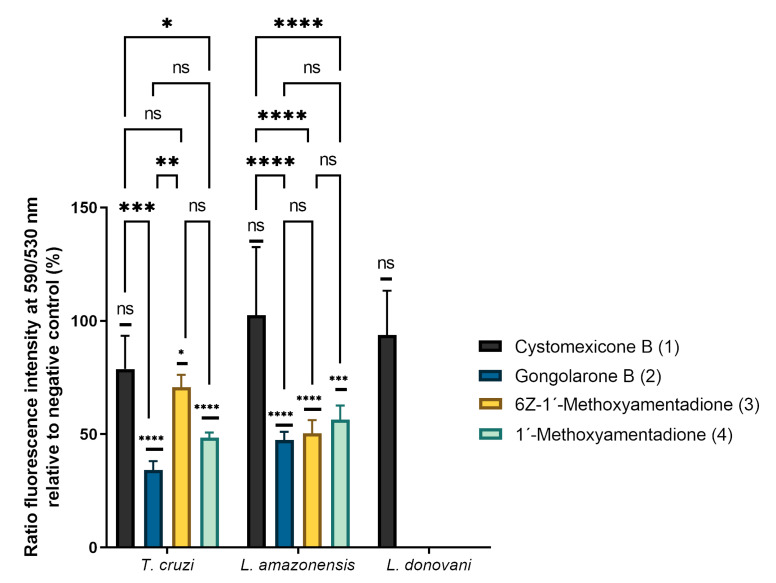
Percentages of ratio fluorescence intensity at 590/530 nm relative to negative control. The analysis of variance was determined by one-way ANOVA using GraphPad.PRISM^®^ 9.0 software. Significance differences when comparing different percentages values are represented like NS non-significant; * *p* < 0.1; ** *p* < 0.01; *** *p* < 0.001 and **** *p* < 0.0001.

**Figure 5 pharmaceuticals-16-00476-f005:**
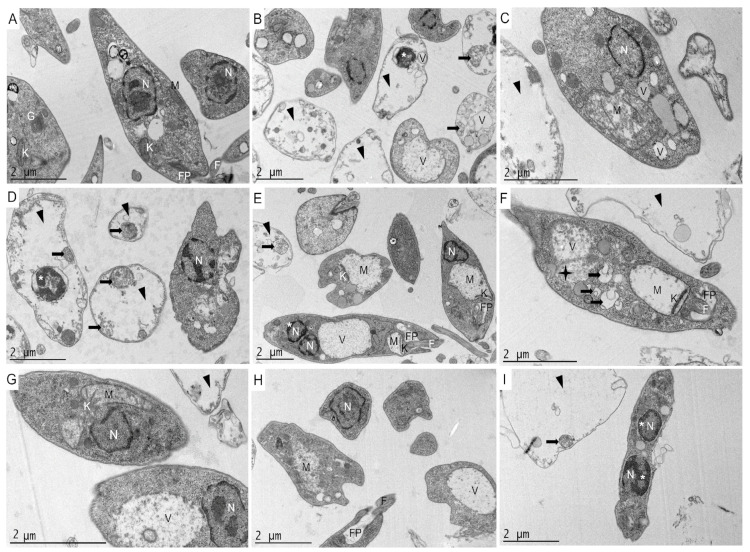
Transmission electron microscopy of *L. amazonensis* promastigotes after 24 h of incubation at the IC_90_ levels of compounds **2**–**4**. The control parasite (image (**A**)) showed its characteristic elongated body and normal morphology of the nucleus (N), mitochondrion (M), kinetoplast (K), Golgi (G), flagellum (F) and flagellar pocket (FP). Similar ultrastructural damage can be observed in *L. amazonensis* promastigotes after incubation with gongolarone B (images (**B**,**C**)), 6*Z*-1′-methoxyamentadione (images (**D**–**F**)) and 1′-methoxyamentadione (images (**G**–**I**)). These compounds caused mitochondrial expansion (M), chromatin condensation in the nucleus (*), intense vacuolization (V), the formation of some autophagy vacuoles (black arrows) and alterations in the morphology of the flagellar pocket (FP). In addition, abundant parasite debris (black triangles) can be observed, confirming the antikinetoplastid activity of these compounds.

**Figure 6 pharmaceuticals-16-00476-f006:**
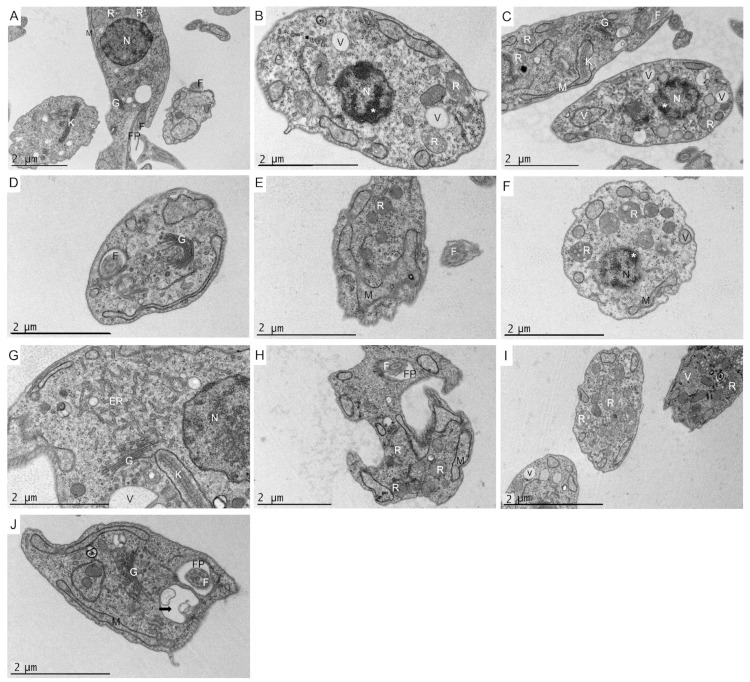
Transmission electron microscopy of *T. cruzi* epimastigotes after 24 h of incubation at the IC_90_ levels of gongolarone B, 6*Z*-1′-methoxyamentadione and 1′-methoxyamentadione. The control parasite (image (**A**)) showed its characteristic elongated body and normal morphology of the nucleus (N), mitochondrion (M), kinetoplast (K), Golgi (G), reservosomes (R), flagellum (F) and flagellar pocket (FP). Similar ultrastructural damage and a discontinuity of the parasite cytoplasm can be observed in *T. cruzi* epimastigotes after incubation with gongolarone B (images (**B**–**D**)), 6*Z*-1′-methoxyamentadione (images (**E**–**G**)) and 1′-methoxyamentadione (images (**H**–**J**)). These compounds caused morphological mitochondrial (M) and kinetoplast (K) alterations, chromatin condensation in the nucleus (*), intense vacuolization (V) and the formation of some autophagy vacuoles (black arrows). All three compounds caused an increase in reservosomes in the cytoplasm. The Golgi (G) and endoplasmic reticulum (ER) were also found to be overdeveloped (with more vacuoles and cisternae than the control). The characteristic morphology and size of the epimastigotes was also altered after incubation with the pure compounds.

**Figure 7 pharmaceuticals-16-00476-f007:**
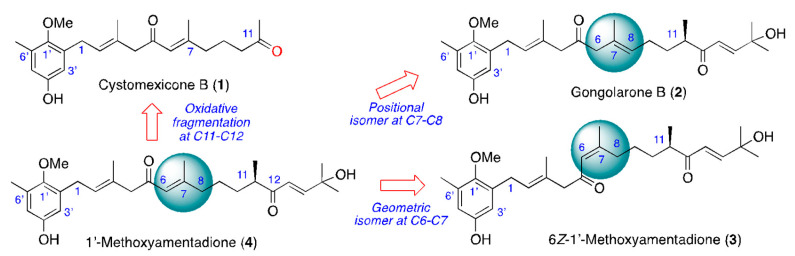
A structure–activity relationship analysis of the antikinetoplastid activity of compounds **1**–**4**.

**Table 1 pharmaceuticals-16-00476-t001:** Antikinetoplastid activity (IC_50_, µM) and cytotoxicity (CC_50_, µM) against eukaryotic cells of compounds **1**–**4** isolated from *G. abies-marina*. ^a^ Selectivity index (CC_50_/IC_50_); * reference drugs; ± represents the standard deviation.

Compounds	*L. amazonensis*	*L. donovani*	*T. cruzi*	Murine Macrophages
Cystomexicone B (**1**)	36.65 ± 4.24 (7.61) ^a^	248.27 ± 0.94 (1.12) ^a^	19.19 ± 2.62 (14.53) ^a^	>278.96
Gongolarone B (**2**)	6.45 ± 0.44 (3.79) ^a^	-	2.92 ± 0.25 (8.36) ^a^	24.41 ± 3.98
6*Z*-1′-Methoxyamentadione (**3**)	4.88 ± 0.33 (5.49) ^a^	97.94 ± 5.32 (0.27) ^a^	3.26 ± 0.11 (8.19) ^a^	26.80 ± 4.06
1′-Methoxyamentadione (**4**)	4.88 ± 0.86 (5.41) ^a^	126.51 ± 1.15 (0.21) ^a^	4.93 ± 0.55 (5.36) ^a^	26.42 ± 2.27
Miltefosine *	6.48 ± 0.10 (11.10) ^a^	3.31 ± 0.11 (21.80) ^a^	-	72.18 ± 1.25
Benznidazole *	-	-	6.95 ± 0.50 (57.51) ^a^	399.91 ± 1.04

**Table 2 pharmaceuticals-16-00476-t002:** Activity against amastigotes of *L. amazonensis* (IC_50_, µM) of compounds **1**–**4**. ^a^ Selectivity index (CC_50_/IC_50_); * reference drugs; ± represents the standard deviation.

Compounds	*L. amazonensis*
Cystomexicone B (**1**)	24.55 ± 0.83 (11.36) ^a^
Gongolarone B (**2**)	16.23 ± 1.90 (1.50) ^a^
6*Z*-1′-Methoxyamentadione (**3**)	16.17 ± 2.61 (1.66) ^a^
1′-Methoxyamentadione (**4**)	27.59 ± 1.68 (0.96) ^a^
Miltefosine *	3.12 ± 0.12 (23.16) ^a^

## Data Availability

The data presented in this study are available on request from the corresponding author.

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
