# Peer review of "Meroterpenoids from Gongolaria abies-marina against Kinetoplastids: In Vitro Activity and Programmed Cell Death Study"

_pharmaceuticals, 2023, doi:10.3390/ph16040476_

Round 1

Reviewer 1 Report

In this article, authors describe Meroterpenoids from Gongolaria abies-marina against kinetoplastids: In vitro activity and programmed cell death study. In my opinion, some issues should be further address and I hope following comments could be helpful for improving their paper:

1. kindly add the main results in the abstract and also add graphical abstract in this manuscript. 

2. why isolated compounds 1-4 were determined against L. amazonensis, L. donovani and T. cruzi.? justify it

3. what kind of murine macrophages was used in this study?

4. why T. cruzi, gongolarone produce the greatest decrease in mitochondrial membrane potential.? 

5. Discussion part needs to revise and compare your results with already published literature to support your hypothesis.

6. Please revisit the entire manuscript for minor grammar and typo issues. 

Author Response

Reviewer 1:
In this article, authors describe Meroterpenoids from Gongolaria abies-marina against
kinetoplastids: In vitro activity and programmed cell death study. In my opinion, some
issues should be further address and I hope following comments could be helpful for
improving their paper:
Thank you very much for your positive feedback and your observations. Please find
below detailed point by point responses to your comments along with the references to
the manuscript. The changes implemented have been highlighted in yellow.
1. kindly add the main results in the abstract and also add graphical abstract in this
manuscript. 
Thank you so much for your observation. We have taken it into account and added the
main results in the abstract [Lines 39-42]. The graphical abstract has been attached to
the manuscript.
2. why isolated compounds 1-4 were determined against L. amazonensis, L. donovani
and T. cruzi.? justify it
Thank you very much for your question. We found it really interesting. Our research
group is working on chemotherapy against different protozoa and during our experience
we have observed the promising leishmanicidal and trypanocidal activity of compounds
of marine origin, such as some compounds from algae (see references below).
Therefore, different species of Leishmania and Trypanosoma cruzi have been used in
this study. Regarding the Leishmania species chosen for these assays, we have selected
species that cause different clinical forms of the disease. L. donovani causes visceral
leishmaniasis and is distributed in the Old World. On the other hand, L. amazonensis
causes the cutaneous form of leishmaniasis and is distributed in the New World. With
these two species we can cover different clinical forms of the disease and evaluate the
therapeutic potential of meroterpenoids 1-4 in both cases. As for T. cruzi, this parasite
belongs to the same family as Leishmania species. As they are closely related, it is
possible that they share some targets on which the isolated compounds may act and
therefore also present sensitivity to these compounds.
REFERENCES:
Chiboub, O.; Sifaoui, I.; Lorenzo-Morales, J.; Abderrabba, M.; Mejri, M.; Fernández,
J.J.; Piñero, J.E.; Díaz-Marrero, A.R. Spiralyde A, an antikinetoplastid dolabellane from
the brown alga Dictyota spiralis. Mar. Drugs. 2019, 17(3),192. doi:
10.3390/md17030192.
Díaz-Marrero, A.R.; López-Arencibia, A.; Bethencout-Estrella, C.J.; Cen-Pacheco, F.;
Sifaoui, I.; Hernández Creus, A.; Duque-Ramírez, M.C.; Souto, M.L.; Hernández
Daranas, A.; Lorenzo-Morales, J.; Piñero, J.E.; Fernández, J.J. Antiprotozoal activities

of marine polyether triterpenoids. Bioorg Chem. 2019, 2, 103276. doi:
10.1016/j.bioorg.2019.103276.
Bethencourt-Estrella, C.J.; Nocchi, N.; López-Arencibia, A.; San Nicolás-Hernández,
D.; Souto, M.L.; Suárez-Gómez, B.; Díaz-Marrero, A.R.; Fernández, J.J.; Lorenzo-
Morales, J.; Piñero, J.E. Antikinetoplastid activity of sesquiterpenes isolated from the
Zoanthid Palythoa aff. clavata. Pharmaceuticals 2021, 14(11), 1095. doi:
10.3390/ph14111095.
3. what kind of murine macrophages was used in this study?
Thank you for your question. In this study we used a murine macrophage cell line
J774A.1 belonging to the ATCC® TIB-67TM strain acquired from the American Type
Culture Collection (ATTC, LGC Promochem, Barcelona, Spain).
4. why T. cruzi, gongolarone produce the greatest decrease in mitochondrial membrane
potential.? 
Thank you very much for your question, we found it a very interesting observation. In
particular, we do not know exactly what mechanism induces this great decrease in
mitochondrial membrane potential in T. cruzi. However, as can be seen in the
experiments carried out, gongolarone B is the compound with the highest trypanocidal
activity. Moreover, as can be seen in the TEM images (Figure 6C), the IC 90 of
gongolarone B causes a deformation of the kinetoplast located in the mitochondria,
which could be related to the decrease of the mitochondrial membrane potential.
5. Discussion part needs to revise and compare your results with already published
literature to support your hypothesis.
Thank you for your comment, we have considered your advice and we have revised and
compared the results with already publishe literatura in the discusión [Lines 343-344;
355-360; 395-406; 428-431; 471-475].
6. Please revisit the entire manuscript for minor grammar and typo issues. 
Thank you so much for your comment,we did not realize this mistakes. We have
carefully revised the manuscript and corrected minor grammar and typo issues.

Reviewer 2 Report

Dear authors

Th MS entitled “Meroterpenoids from Gongolaria abies-marina against kinetoplastids: In vitro activity and programmed cell death study” was thoroughly reviewed. The MS is comprehensive but well elaborated. Abstract is concise however, lacking some experimentally obtained values. The introduction is sufficient and provide valid research questions. The aims of this study are clear. The results are promising and a detail insight has been provided by the authors. The figures and pictorial representations are good. the discussion seems to be lengthy however, worthy. the procedures for isolation are satisfactory. The conclusion is in line with the proposed and obtained outcomes.

My comments are:

1. kindly provide the structural data for each of the compound in experimental sections.

2: Also provide details of the physical data of each compound.

Author Response

Reviewer 2: My comments are: Thank you very much for your feedback. Please find below detailed point by point responses to your comments along with the references to the manuscript. The changes implemented have been highlighted in yellow. 1. kindly provide the structural data for each of the compound in experimental sections. Thank you very much for your comment. We have taken it into account and added this structural data of the compounds isolated [Lines 525-561]. 2: Also provide details of the physical data of each compound. Thank you so much for your observation. We added the physical data of the compounds in experimental section [Lines 525-561].

Reviewer 3 Report

This is an excellent work on the biological importance of marine natural materials.

My questions and suggestions are as follows

1. A brief overview would be good for the introduction: what meroterpenoids are.

2. Figure 1: lacks compound numbering.

3. The structural formula of Miltefosine and Benznidazole should be presented.

4. Figures 4-16 should be moved to the Supplement chapter and only a detailed discussion would remain in the text.

Author Response

Reviewer 3: My questions and suggestions are as follows Thank you very much for your feedback. Please find below detailed point by point responses to your comments along with the references to the manuscript. The changes implemented have been highlighted in yellow. 1. A brief overview would be good for the introduction: what meroterpenoids are. Thank you so much for your advice. We have taken it into account and we added a description of meroterpenoids in the introduction of the manuscript [Lines 109-116]. 2. Figure 1: lacks compound numbering. Thank you for your observation. We have changed figure 1 following your recomendation [Line 127]. 3. The structural formula of Miltefosine and Benznidazole should be presented. Thank you so much for your comment. We have taken it into account and we have added the structural formula of miltefosine and benznidazol [Lines 163]. 4. Figures 4-16 should be moved to the Supplement chapter and only a detailed discussion would remain in the text. Thank you very much for your observation. We have considered your advice and therefore we have moved these figures to the supplement chapter.